# Effect of Gap Size and Elevation on the Regeneration and Coexistence of *Abies*, *Betula*, and *Acer* Tree Species in a Subalpine Coniferous Forest

Wangya Han [1,2,*] , Li Chen [2,3], Jingyang Liu [4], G. Geoff Wang [5], Dan Liu [6] and Guohua Liu [2,7,*]

1 Institute of Ecology, School of Applied Meteorology, Nanjing University of Information Science & Technology, Nanjing 210044, China
2 State Key Laboratory of Urban and Regional Ecology, Research Center for Eco-Environmental Sciences, Chinese Academy of Sciences, Beijing 100085, China; ctpchenlly@163.com
3 Torch High Technology Industry Development Center, Ministry of Science & Technology, Beijing 100045, China
4 Reading Academy, Nanjing University of Information Science & Technology, Nanjing 210044, China
5 Department of Forestry and Environmental Conservation, Clemson University, Clemson, SC 29634, USA
6 Institute of Qinghai-Tibetan Plateau, Southwest Minzu University, Chengdu 610041, China
7 College of Resources and Environment, University of Chinese Academy of Sciences, Beijing 100049, China
* Correspondence: wyhan@nuist.edu.cn (W.H.); ghliu@rcees.ac.cn (G.L.)

**Abstract:** Forest gaps play an important role in species regeneration and forest succession. Gap size has a primary influence on tree species coexistence and community assembly along an elevation gradient. In this study, we evaluated the regeneration and coexistence of *Abies faxoniana*, *Betula utilis*, and *Acer maximowiczii* at different life history stages in varied gap sizes along an elevation gradient (between 3000 and 3500 m a.s.l). We found that gap size can positively enlarge the effects of elevation on the regeneration density of the three species. In the process of regeneration from seedling to sapling, *Abies* had stronger regeneration capability, with regeneration niche breadths of more than 0.6 in different gap sizes. A factor analysis of mixed data indicated that regeneration density, soil nutrient contents, and air humidity were mainly related to gap size, but habitat temperature was largely determined by elevation. The connection between the species regeneration density and habitat conditions was due to differences in their regeneration niches, reflecting the selective preemption of environmental resources of different species in different life stages. Microhabitat heterogeneity, controlled by the characteristics of forest gaps along an elevation, affected the regeneration niche differences of the tree species, which contributed to the species coexistence and community assembly processes.

**Keywords:** gap size; species regeneration; species coexistence; community assembly; subalpine coniferous forest





## 1. Introduction

Forest gaps increase the heterogeneity of environmental conditions in forests and promote the regeneration and coexistence of species in the forest understory [1]. Gap formation is an effective silvicultural approach to achieving continuous forest regeneration and succession [2]. Compared with traditional silviculture policy, this close-to-nature type of forest management highlights the importance of gap disturbance [3]. Gap size, as an important gap feature reflecting the degree of disturbance, can strongly influence the microclimate and therefore species regeneration and other ecological processes [1,4]. Thus, understanding how exactly gap size affects species regeneration and coexistence has been an important topic in forest ecology and management [5–7].

Most studies examining canopy gaps focused on gap regeneration on temporal scales [8,9]. However, little attention has been paid to gap regeneration on spatial scales, especially along an elevation gradient [4]. As an important geographical factor, elevation

is a strong driver of spatial variation in environmental factors (e.g., temperature, precipitation, and soil properties) that affect species distribution, species regeneration, and coexistence [10,11]. For example, a study on canopy gaps in Atlantic Rainforests suggested that a low altitude (at approximately 350 m a.s.l.) was more conducive to plant species regeneration and coexistence [12]. Sun et al. suggested that subalpine *Abies fabri* in mountainous regions was more sensitive to disturbances in air temperature in high-elevation areas than low-elevation areas [13]. In addition, the underground environmental factors of the subalpine forest also exhibited significant differences along an elevation gradient from 2800 to 3600 m [14]. All these studies investigated the effects of elevation on plant species regeneration and environmental factors, but the observed changes in species regeneration and environmental factors could also be related to varied gap sizes along the elevation gradient. Considering the complexity of the ecological process underlying gap regeneration, a combined investigation to better understand species regeneration and coexistence along gradients of gap sizes and elevation may warrant more detailed insights into the underlying processes.

The abundance and distribution of plant species are influenced by their early regeneration stages, that is, the density and traits of seedlings and saplings, which are susceptible to numerous environmental factors [15,16]. The differentiation of the regeneration niche between different life history stages is one key to explain species regeneration and coexistence in forest communities due to different requirements for the microhabitats of species in different life stages [17]. In the regeneration niches of species, distinct differences occur in different life stages. Studies are needed to understand how changes in the ontogenetic regeneration niche drive spatial distribution and successional dynamics. Variations in the regeneration density of seedlings, saplings, and small trees are linked to different regeneration niches and species' responses to their microenvironment [18,19]. Environmental resource gradients cause interspecific differences in the survival and growth of seedlings and saplings. Studies have shown that the interspecific tradeoffs between seedlings and saplings restrict species to certain succession gradients, which can explain community assembly and succession dynamics [20,21]. Moreover, regeneration density is an important factor in sustainable forest management, and density dependence has been hypothesized as one of the most prominent mechanisms in species coexistence [22,23]. Consequently, analyzing the association between species regeneration and the environment can better explain the species coexistence mechanism and gap species regeneration, providing an appropriate forest management strategy.

In China's subalpine coniferous forests, the forests have mixed canopies dominated by coniferous species. Historically, the old-growth forest canopy in the Wolong National Nature Reserve was dominated by *Abies* species at altitudes of more than 3000 m [4,24]. A recent study showed that broadleaf species (e.g., *Betula*) have increased in the subalpine coniferous forest with canopy disturbance [25]. Species regeneration and community succession are related to species coexistence and community assembly [17,26]. At present, *Abies*, *Betula*, and *Acer* are major tree species in most subalpine coniferous gaps from 3000 to 3500 m in the Wolong National Nature Reserve. To explore how the microhabitat conditions of different species affect species regeneration along the elevation gradient in gaps, we aimed to answer the following questions: (1) How does gap size influence species regeneration? (2) How do gap size and elevation affect species regeneration and environmental factors? (3) What are the relationships between species regeneration and microhabitat environmental factors?

## 2. Materials and Methods

### 2.1. Study Site

The subalpine coniferous forest in this study was an old-growth forest on Mount Nadu in the Wolong Nature Reserve (30°45′–31°25′ N, 102°52′–103°24′ E), Sichuan Province, Southwest China (Figure 1). The study area has a subtropical inland and mountainous climate with strong solar radiation and cold temperatures. The mean annual temperature

is between 4.1 and 5.1 °C. The mean annual rainfall is between 861 and 1800 mm, with the highest precipitation month, August, averaging 340 mm. The main soil type is brown coniferous forest soil, followed by subalpine meadow soil. The parent rock of these two soil types is weathered killas [4]. Most of the snowfall occurs from October to February at high elevations. More detailed information on the study area can be found in earlier publications [4,27,28].

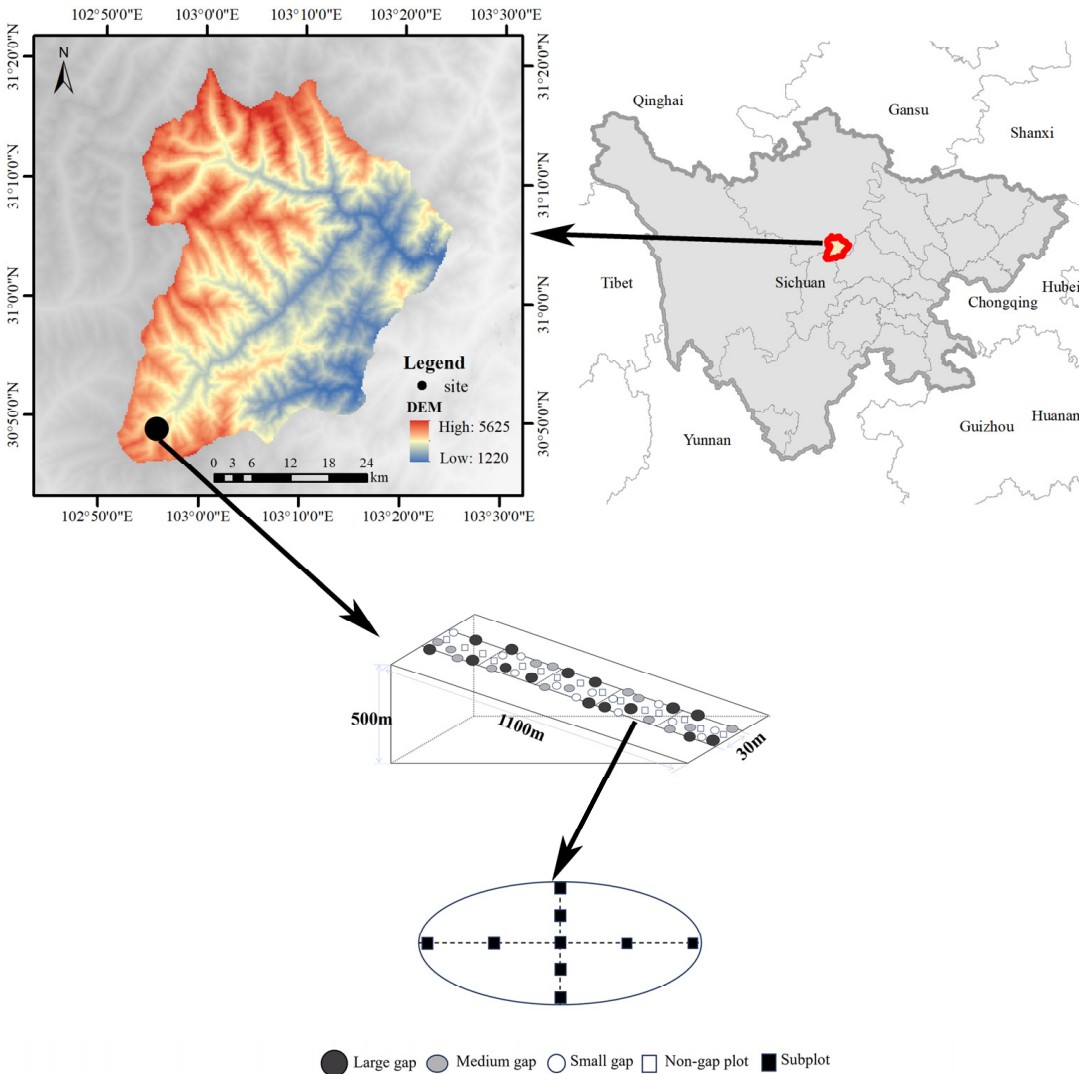

**Figure 1.** Location, sampling plots, and quadrats in the Wolong National Nature Reserve in Southwest China.

Our study was conducted in a coniferous forest with little human disturbance located at elevations between 3000 and 3500 m. Snowstorms and other natural disturbances are the main causes of forest gap formation. In this old-growth forest, the conifer tree *Abies faxoniana* is mostly found from 2000 to 3800 m and is considered a shade-tolerant species. In the 3000 to 3500 m elevation range, the subalpine coniferous forest also includes *Betula utilis* and *Acer maximowiczii*, two broadleaf tree species. These three dominant tree species comprise 95% of all trees at this range of elevation. The shrub species mainly include *Rhododendron faberisp*, *Rhododendron phaeochrysum*, and *Fargesia nitida*. The herb layer is dominated by *Circaea alpine*, *Oxalis acetosella*, and *Parasenecio deltophyllus*.

*2.2. Gap Selection and Experimental Design*

A randomized design was used to sample gaps in each 100 m elevation interval between 3000 and 3500 m during the 2015–2018 period. At each elevation level, three gaps in each size class and a control site with no gap were selected. The three gap sizes were classified as small ($n$ = 3, area range: 38.5–100 m$^2$), medium ($n$ = 3, area range: 101–200 m$^2$), and large ($n$ = 3, area range: 201–402 m$^2$) and were determined to be expanded gaps [2]. The non-gap was set up in the forest understory as a control ($n$ = 3, area range: 10 × 10 m). In total, we sampled 60 plots with similar forest gap ages for five elevation levels from 3000 to 3500 m. We measured the longest canopy opening dimension as the gap length and the distance perpendicular to the gap length across the center of the gap as the gap width, and we then used these data to calculate the gap area using the formula $A = \pi ab$, with $a$ and $b$ representing half of the gap length and width if the gap was of a more oval shape. If the gap was more circular, the formula $A = \pi r^2$ was used, with $r = (a + b)/2$. The non-gap plots were located at least 10 m away from the plots with gaps. In each plot, nine quadrats (1 × 1 m) were set up using the cross-line method and radiated outward from east to west and from north to south, with one located at the center of the plot and the others distributed along the two line transects [28].

*2.3. Data Collection*

2.3.1. Regeneration Characteristics

Within each gap, all seedlings, saplings, and small trees of the three tree species (*Abies*, *Betula*, and *Acer*) were investigated during the growing season, and their species, height, and ground line diameter (GLD) were recorded. Considering the difference in the growth of the three species, we divided them into three different regeneration stages (seedlings, saplings, and small trees) based on our previous studies [4]. The seedling stage was defined as "0 < stems ≤ 0.1 m in height or 0 < GLD ≤ 0.1 cm". The sapling stage was defined as "0.1 m < stems < 1 m in height and 0.1 cm < GLD < 1.5 cm" [29]. The regeneration stage of the small trees was defined as "either stems ≥ 1 m in height or GLD ≥ 1.5 cm". Stems with a height > 5 m or a diameter at breast height (DBH) > 10 cm were removed from the regeneration datasets because they were considered mature. The regeneration capacities were weaker in small gaps and at high elevation gaps, especially for *Betula* and *Acer*.

The importance value (IV) of each species was calculated as follows:

$$P_i = \left( \frac{D_i}{\sum_i^S D_i} + \frac{G_i}{\sum_i^S G_i} + \frac{F_i}{\sum_i^S F_i} \right) / 3 \times 100 \tag{1}$$

where $P_i$ is the importance value (IV) of species $i$; $D_i$ is the density of species $i$; $G_i$ is the GLD of species $i$; and $F_i$ is the frequency of species $i$ [30].

Niche breadth, regeneration niche breadth, and regeneration niche overlap were used to assess the niche characteristics of the tree species: Levin's niche breadth index ($B_L$) [31]; the regeneration niche breadth [32]; the niche overlap value [31]; and the regeneration niche overlap value [32]. The equations of each index were calculated as follows:

$$B_{Li} = 1 / \left( n \sum_{j=1}^r P_{ij}^2 \right) \tag{2}$$

$$B_i' = (B_{ib} + A_i B_{ia}) / \sum (B_{jb} + A_j B_{ia}) \tag{3}$$

$$O_{ij} = \sum_{j=1}^r P_{ik} \frac{P_{jk}}{\sqrt{\sum P_{ik}^2 \sum P_{jk}^2}} \tag{4}$$

$$O_{ij}' = (O_{S1} + O_{S2}) / 2 \tag{5}$$

where $B_{Li}$ is the Levin's niche breadth index ($B_L$) value of species $i$; $P_{ij}$ is the proportion of the dominance of the $i^{th}$ population in the $j^{th}$ regeneration stage to the total dominance of the $i^{th}$ population in all regeneration stages; $n$ is the number of regeneration stages; $B_i'$ is the regeneration niche breadth ($B_i'$) of species $i$; $B_{ia}$ and $B_{jb}$ are the niche breadths of species $i$ at stage $a$ and species $j$ at stage $b$; $O_{ij}$ is the niche overlap value between species $i$ and $j$; $P_{ik}$ is the proportion of the dominance of species $i$ in the regeneration stage $k$; $P_{jk}$ is the proportion of the dominance of species $j$ in the regeneration stage $k$; $O_{ij}'$ is the regeneration niche overlap value; and $O_{S1}$ and $O_{S2}$ represent the niche overlap values between species $i$ and $j$ in different regeneration stages and can be calculated using Equation (4).

### 2.3.2. Environmental factors

We measured the solar radiation, air temperature, air humidity, soil temperature, and soil nutrients in each plot along the elevation gradient. The photosynthetic photon flux density (LI-190, Li-Cor Inc., Lincoln, NE, USA) was measured at 1 m above ground level. Within each plot, the mean daily PPFD ($\mu$ mol m$^{-2}$ s$^{-1}$) datasets were recorded continuously as 2 h averages from 8:00 a.m. to 6:00 p.m. on cloudless days and overcast days during the growing season (June–September 2015–2018) [33]. We measured the annual mean air temperature (AMAT, °C), air relative humidity (AH, %), and soil temperature (ST, °C) [4]. We also measured and calculated the annual average temperature difference between day and night (DAMT, °C), the annual mean temperature of the growing/non-growing season (GST/NGST, °C), and the soil temperature in the upper (0–10 cm, ST$_1$, °C) and deep (10–20 cm, ST$_2$, °C) layers.

These datasets were measured using a button thermometer (iButton DS1923-F5, Maxim/Dallas Semiconductor, Sunnyvale, CA, USA), and data were recorded every 2 h for one year at least. For the above-ground environmental factors, two iButton recorders were fixed separately in the center and at the edge of each plot. The soil temperature datasets were monitored in the center of each plot at depths of 0–10 cm and 10–20 cm. Three random soil samples were collected beneath the litter layer in each plot. All soil samples were air-dried at room temperature. Stones, roots, and debris were removed, and the samples were sieved through a 2 mm mesh screen [28]. The soil total carbon (STC, %), soil total nitrogen (STN, %), and the carbon-to-nitrogen ratio (C/N, %) were measured in each sampling quadrat using a Vario EL III analyzer (Elementar Analysensysteme GmbH, Hanau, Germany).

### 2.4. Statistical Analysis

The regeneration data of all plots were averaged in each gap before analysis. We used a one-way ANOVA to test the difference in the regeneration density among the three tree species in the same gap size and elevation gradient. A two-way ANOVA was used to test the effects of gap size and elevation on the regeneration density of seedlings, saplings, and small trees from the three species. Two-way ANOVA was used to test the effects of gap size and elevation on the environmental factors. Each $p$-value less than 0.05 was regarded as statistically significant. Tukey's post hoc tests were used to further examine the differences among treatment levels. The Bonferroni adjustment test was applied for multiple testing, and a $p$-value less than 0.01 indicated a significant difference. Pearson's correlation coefficient was calculated to assess the correlation between regeneration density and environmental factors. A factor analysis of mixed data (FAMD), a mix between a principal component analysis (PCA) and a multiple correspondence analysis (MCA), was used to assess the relationship between regeneration density and environmental factors. Data were analyzed using R version 3.3.3 (R Core Team, Vienna, Austria, 2017). The multcomp package was used to conduct the descriptive statistics, one-way ANOVA, two-way ANOVA, and Tukey's post hoc tests. The psych package was used to conduct the correlation analyses. The FactoMineR and factoextra packages were used to conduct the FAMD.

## 3. Results

### 3.1. Variations in the Regeneration Density and Niche of the Three Species among Gaps

Seedlings had the highest regeneration density, higher than saplings and small trees within each gap microhabitat (Figure 2, Table 1). As expected, the regeneration densities of the seedlings, saplings, and small trees all increased when the gap size increased (Figure 2). The seedling densities of the three species were significantly higher in gap habitats compared to the understory (Table 1). In small gaps, the *Abies* seedling density was significantly higher than the *Betula* and *Acer* seedling densities, and the *Betula* seedling density was significantly higher than the *Acer* seedling density (Figure 2). The *Abies* sapling density was significantly greater than the *Betula* and *Acer* sapling densities, but the difference between the *Betula* and *Acer* sapling densities was not significant. In medium and large gaps, the sapling and small tree densities of *Betula* were significantly higher than the densities of *Abies* and *Acer*. In large gaps, the seedling, sapling, and small tree densities of *Betula* were significantly higher than those of *Abies*. These results indicated that small gaps were conducive to the regeneration of the *Abies* saplings, and large gaps facilitated the regeneration of *Betula*.

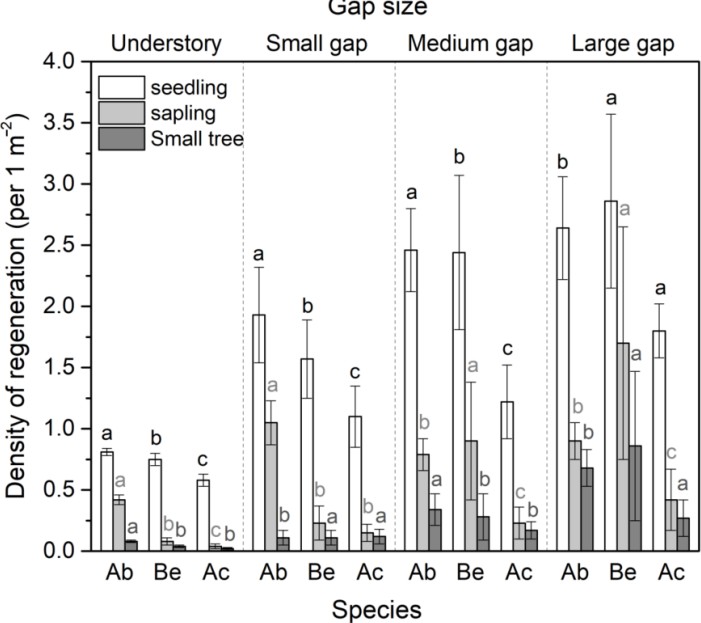

**Figure 2.** ANOVA analysis of the three species' seedling, sapling, and small-tree regeneration densities in different gap sizes and the understory. Different letters indicate significant differences ($p < 0.05$). Error bars represent one standard error. Ab: *Abies faxoniana*; Be: *Betula utilis*; Ac: *Acer maximowi*.

The niche breadth of *Acer* in the understory was broader than that of the other two species at the seedling, sapling, and small-tree stages (Table 1). Across different gap sizes, the niche breadths of the three species had similar trends, with the narrowest niche breadth occurring for *Betula* in the sapling stage (Table 1). For the seedling-to-sapling regeneration process, *Abies* had a stronger regeneration capability, with regeneration niche breadths of more than 0.6 in gaps of different sizes. The regeneration niche overlap of *Abies* and *Acer* was the largest among species pairs in both the seedling-to-sapling and sapling-to-small tree regeneration processes (Table 2). This indicates that competition and coexistence exist between *Abies* and *Acer*, and *Abies* has a competitive advantage.

**Table 1.** Regeneration density and niche breadth values of the three species among gaps. Different letters indicate significant differences ($p < 0.05$).

|  |  | LG | MG | SG | Us | Total |
|---|---|---|---|---|---|---|
| **Regeneration density** |  |  |  |  |  |  |
| *Abies faxoniana* | Seedling | 2.64 ± 0.42 a | 2.46 ± 0.34 a | 1.93 ± 0.39 b | 0.81 ± 0.03 c | 1.96 ± 0.79 |
|  | Sapling | 0.90 ± 0.15 b | 0.79 ± 0.13 b | 1.05 ± 0.18 a | 0.42 ± 0.04 c | 0.79 ± 0.27 |
|  | Small tree | 0.68 ± 0.15 a | 0.34 ± 0.13 b | 0.11 ± 0.06 c | 0.08 ± 0.01 c | 0.30 ± 0.26 |
| *Betula utilis* | Seedling | 2.86 ± 0.71 a | 2.44 ± 0.63 a | 1.57 ± 0.32 b | 0.75 ± 0.05 c | 1.90 ± 0.96 |
|  | Sapling | 1.70 ± 0.95 a | 0.90 ± 0.48 b | 0.23 ± 0.14 c | 0.08 ± 0.03 c | 0.73 ± 0.83 |
|  | Small tree | 0.86 ± 0.61 a | 0.28 ± 0.19 b | 0.11 ± 0.06 b | 0.04 ± 0.01 c | 0.32 ± 0.45 |
| *Acer maximowiczii* | Seedling | 1.80 ± 0.22 a | 1.22 ± 0.30 b | 1.10 ± 0.25 b | 0.58 ± 0.05 c | 1.18 ± 0.49 |
|  | Sapling | 0.42 ± 0.25 a | 0.23 ± 0.13 b | 0.15 ± 0.07 b c | 0.04 ± 0.02 c | 0.21 ± 0.20 |
|  | Small tree | 0.27 ± 0.15 a | 0.17 ± 0.07 b | 0.12 ± 0.06 b | 0.02 ± 0.01 c | 0.14 ± 0.13 |
| **VI** |  |  |  |  |  |  |
| *Abies faxoniana* | Seedling | 4.087 | 4.421 | 5.224 | 2.572 | 16.306 |
|  | Sapling | 2.861 | 3.631 | 3.882 | 2.705 | 13.079 |
|  | Small tree | 3.466 | 3.767 | 4.144 | 2.827 | 14.205 |
| *Betula utilis* | Seedling | 3.202 | 4.130 | 5.022 | 2.124 | 14.477 |
|  | Sapling | 2.346 | 3.184 | 3.697 | 1.214 | 10.441 |
|  | Small tree | 2.314 | 3.276 | 3.795 | 2.569 | 11.954 |
| *Acer maximowiczii* | Seedling | 1.685 | 2.020 | 2.173 | 1.004 | 6.883 |
|  | Sapling | 0.861 | 1.215 | 1.441 | 0.893 | 4.410 |
|  | Small tree | 1.860 | 2.161 | 2.435 | 1.789 | 8.244 |
| **Niche breadth** |  |  |  |  |  |  |
| *Abies faxoniana* | Seedling | 0.989 | 0.958 | 0.987 | 0.989 | 0.969 |
|  | Sapling | 0.988 | 0.990 | 0.997 | 0.887 | 0.900 |
|  | Small tree | 0.988 | 0.975 | 0.973 | 0.813 | 0.918 |
| *Betula utilis* | Seedling | 0.946 | 0.905 | 0.940 | 1.325 | 0.977 |
|  | Sapling | 0.889 | 0.765 | 0.869 | 3.999 | 0.811 |
|  | Small tree | 0.989 | 0.897 | 0.934 | 0.990 | 0.881 |
| *Acer maximowiczii* | Seedling | 0.927 | 0.906 | 0.954 | 6.424 | 0.969 |
|  | Sapling | 0.990 | 0.888 | 0.919 | 8.198 | 0.891 |
|  | Small tree | 0.997 | 0.988 | 0.985 | 2.034 | 0.976 |
| **Regeneration niche breadth** |  |  |  |  |  |  |
| *Abies faxoniana* | Seedling |  |  |  |  |  |
|  | Sapling | 0.612 | 0.662 | 0.611 | 0.105 | 0.651 |
|  | Small tree | 0.575 | 0.584 | 0.559 | 0.179 | 0.584 |
| *Betula utilis* | Seedling |  |  |  |  |  |
|  | Sapling | 0.487 | 0.459 | 0.503 | 0.400 | 0.478 |
|  | Small tree | 0.470 | 0.454 | 0.487 | 0.462 | 0.457 |
| *Acer maximowiczii* | Seedling |  |  |  |  |  |
|  | Sapling | 0.414 | 0.403 | 0.400 | 1.626 | 0.393 |
|  | Small tree | 0.460 | 0.469 | 0.458 | 1.138 | 0.465 |

**Table 2.** Regeneration niche overlap of the three species. S1: *Abies faxoniana*; S2: *Betula utilis*; S3: *Acer maximowiczii*.

|  | Seedling–Sapling | | | Sapling–Small Tree | | |
|---|---|---|---|---|---|---|
| **Total** |  |  |  |  |  |  |
| Species | S1 | S2 | S3 | S1 | S2 | S3 |
| S1 | 1.00 |  |  | 1.00 |  |  |
| S2 | 0.94 | 1.00 |  | 0.94 | 1.00 |  |
| S3 | 0.98 | 0.97 | 1.00 | 0.98 | 0.94 | 1.00 |
| **Large gap** |  |  |  |  |  |  |
| Species | S1 | S2 | S3 | S1 | S2 | S3 |
| S1 | 1.00 |  |  | 1.00 |  |  |
| S2 | 0.97 | 1.00 |  | 0.98 | 1.00 |  |
| S3 | 0.98 | 0.96 | 1.00 | 0.99 | 0.98 | 1.00 |
| **Medium gap** |  |  |  |  |  |  |
| Species | S1 | S2 | S3 | S1 | S2 | S3 |
| S1 | 1.00 |  |  | 1.00 |  |  |
| S2 | 0.94 | 1.00 |  | 0.94 | 1.00 |  |
| S3 | 0.97 | 0.98 | 1.00 | 0.98 | 0.97 | 1.00 |
| **Small gap** |  |  |  |  |  |  |
| Species | S1 | S2 | S3 | S1 | S2 | S3 |
| S1 | 1.00 |  |  | 1.00 |  |  |
| S2 | 0.96 | 1.00 |  | 0.96 | 1.00 |  |
| S3 | 0.97 | 0.98 | 1.00 | 0.98 | 0.98 | 1.00 |

**Table 2.** *Cont.*

| Understory | Seedling–Sapling | | | Sapling–Small Tree | | |
|---|---|---|---|---|---|---|
| Species | S1 | S2 | S3 | S1 | S2 | S3 |
| S1 | 1.00 | | | 1.00 | | |
| S2 | 0.97 | 1.00 | | 0.98 | 1.00 | |
| S3 | 0.99 | 0.97 | 1.00 | 0.99 | 0.98 | 1.00 |

### 3.2. Effect of Gap Size and Elevation on Species Regeneration and Environmental Factors

Species regeneration was influenced by the interaction between gap size and elevation (Figure 3). In the same elevation gradient, the seedling regeneration densities of the three tree species were positively affected by the gap size. The sapling regeneration density of *Abies* was higher in small gaps than in other gap microhabitats, but the sapling regeneration densities of *Betula* and *Acer* increased with an increase in the gap size. The small-tree regeneration densities of the three tree species increased as the gap sizes increased, whereas the small-tree regeneration density of *Betula* was not significantly affected by gap size at the 3400–3500 m elevation gradient (Figure 3C).

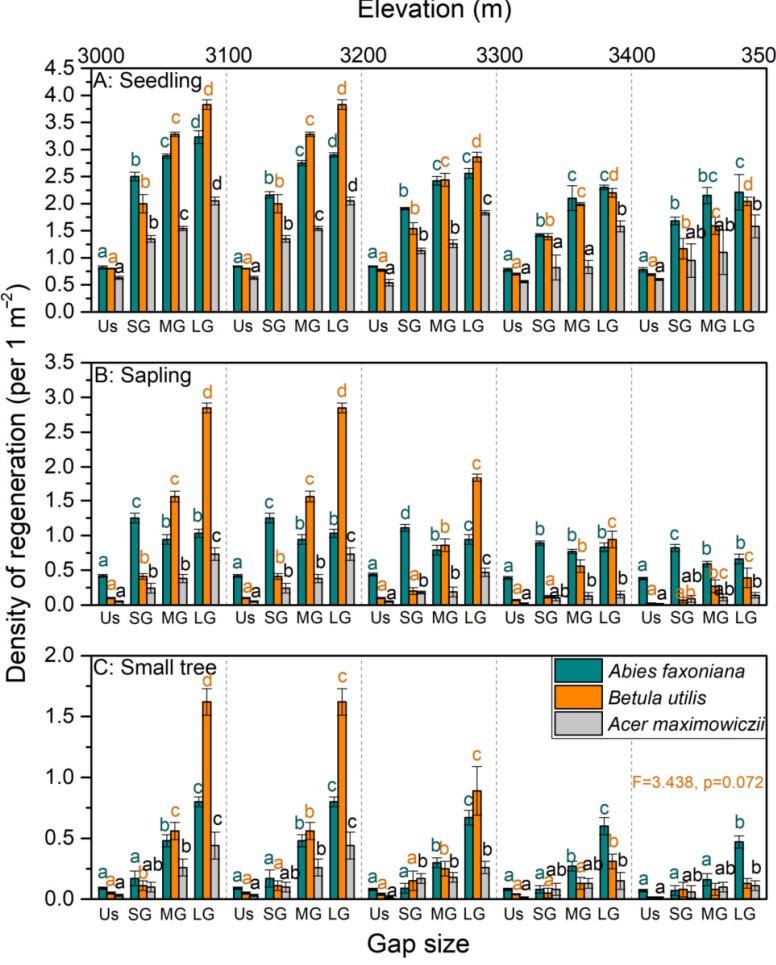

**Figure 3.** ANOVA analysis of the regeneration density values of the seedlings, saplings, and small trees of the three species in gaps of different sizes and the understory at each elevation interval. Different letters indicate significant differences ($p < 0.05$) among treatments.

In the understory, decreases in the regeneration densities of the three species were not obvious with an increasing elevation. In small gaps (Figure 4B₁–B₃), the regeneration density of the *Betula* seedlings was significantly and negatively affected by elevation (F = 17.74,

$p < 0.001$), and the regeneration densities of the *Abies* and *Acer* seedlings decreased significantly at an elevation gradient from 3000 to 3400 m. The sapling regeneration densities of the three species were significantly and negatively affected by the elevation (*Abies*: F = 40.06, $p < 0.001$; *Betula*: F = 29.25, $p < 0.001$; *Acer*: F = 4.48, $p = 0.025$), whereas the small-tree regeneration densities were similar among different elevation gradients. In medium gaps (Figure 4C₁–C₃), the seedling regeneration densities of *Abies* and *Acer* decreased significantly at an elevation gradient from 3000 to 3400 m, and the *Betula* seedling regeneration density was significantly and negatively affected by the elevation (F = 155.74, $p < 0.001$). The sapling and small-tree regeneration densities of the three species were significantly and negatively affected by the elevation (Table 3). In large gaps (Figure 4D₁–D₃), the seedling, sapling, and small-tree regeneration densities of the three species were significantly and negatively affected by elevation, and the effect of elevation on regeneration density was more obvious for *Betula* than for the other two species. There was a significant interaction between gap size and elevation on regeneration density for the seedlings, saplings, and small trees of the three species except for *Acer* seedlings (F = 1.95, $p = 0.057$) (Table 4).

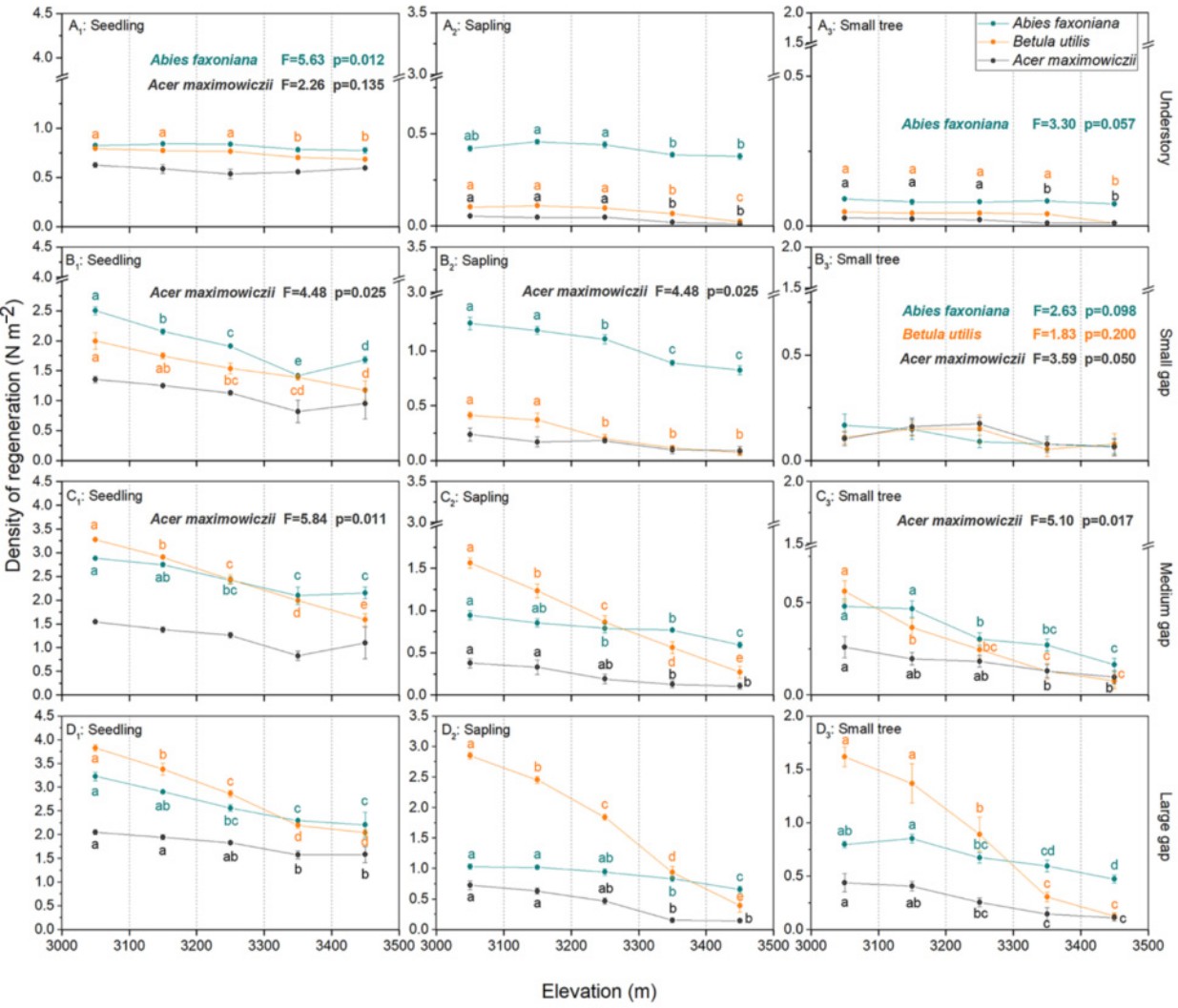

**Figure 4.** ANOVA analysis of the seedling, sapling, and small-tree regeneration densities of the three species in the same gap size along the elevation gradient. Different letters indicate significant differences ($p < 0.05$) among treatments.

**Table 3.** Effects of elevation on the three species' regeneration densities in the same gap size. Significant effects ($p < 0.01$) are bolded.

| Speices | | Large Gap | | Medium Gap | | Small Gap | | Understory | |
|---|---|---|---|---|---|---|---|---|---|
| | | F | $p$ | F | $p$ | F | $p$ | F | $p$ |
| *Abies faxoniana* | seedling | **20.71** | **<0.001** | **21.88** | **<0.001** | **166.61** | **<0.001** | 5.63 | 0.012 |
| | sapling | **22.43** | **<0.001** | **17.22** | **<0.001** | **40.06** | **<0.001** | **10.91** | **<0.001** |
| | small tree | **25.06** | **<0.001** | **25.36** | **<0.001** | 2.63 | 0.098 | 3.30 | 0.057 |
| *Betula utilis* | seedling | **170.88** | **<0.001** | **155.74** | **<0.001** | **17.74** | **<0.001** | **29.19** | **<0.001** |
| | sapling | **339.97** | **<0.001** | **99.05** | **<0.001** | **29.25** | **<0.001** | **57.50** | **<0.001** |
| | small tree | **59.28** | **<0.001** | **34.42** | **<0.001** | 1.83 | 0.200 | **34.17** | **<0.001** |
| *Acer maximowiczii* | seedling | **10.85** | **<0.001** | 5.84 | 0.011 | 4.48 | 0.025 | 2.26 | 0.135 |
| | sapling | **56.30** | **<0.001** | **8.91** | **0.002** | 4.48 | 0.025 | **54.67** | **<0.001** |
| | small tree | **14.31** | **<0.001** | 5.10 | 0.017 | 3.59 | 0.050 | **13.25** | **<0.001** |

**Table 4.** Effects of gap size and elevation on the three species' regeneration densities. Significant effects ($p < 0.01$) are bolded.

| Species | Variables | Seedling | | Sapling | | Small Tree | |
|---|---|---|---|---|---|---|---|
| | | F | $p$ | F | $p$ | F | $p$ |
| *Abies faxoniana* | Gap size | 867.28 | **<0.001** | 483.17 | **<0.001** | 631.14 | **<0.001** |
| | Elevation | 93.20 | **<0.001** | 79.40 | **<0.001** | 45.83 | **<0.001** |
| | Gap size × Elevation | 10.07 | **<0.001** | 7.40 | **<0.001** | 8.78 | **<0.001** |
| *Betula utilis* | Gap size | 1458.34 | **<0.001** | 1654.16 | **<0.001** | 304.98 | **<0.001** |
| | Elevation | 265.93 | **<0.001** | 448.71 | **<0.001** | 83.67 | **<0.001** |
| | Gap size × Elevation | 36.76 | **<0.001** | 122.53 | **<0.001** | 38.62 | **<0.001** |
| *Acer maximowiczii* | Gap size | 176.92 | **<0.001** | 140.60 | **<0.001** | 75.26 | **<0.001** |
| | Elevation | 18.23 | **<0.001** | 56.76 | **<0.001** | 20.37 | **<0.001** |
| | Gap size × Elevation | 1.95 | 0.057 | 13.17 | **<0.001** | 5.89 | **<0.001** |

The photosynthetic photon flux density (PPFD) on overcast days decreased significantly from the large gap to the forest understory ($F = 27.57$, $p < 0.001$) (Table 5 and Table S1). The annual mean temperature of the growing season was not significantly affected by gap size ($F = 2.09$, $p = 0.141$). The annual mean air temperature (AMT), annual average temperature difference between day and night (DAMT), the annual mean temperature of the growing season (GST), and the annual mean temperature of the non-growing season (NGST) decreased significantly an increase in the elevation ($p < 0.001$). The soil temperature (ST) showed no difference among gaps of different sizes (0–10 cm: $F = 0.09$, $p = 0.914$; 10–20 cm: $F = 1.86$, $p = 0.173$), though it decreased significantly as the elevation increased (0–10 cm: $F = 127.26$, $p < 0.001$; 10–20 cm: $F = 106.56$, $p < 0.001$). The soil total carbon (STC) (0–10 cm: $F = 11.68$, $p < 0.001$; 10–20 cm: $F = 371.10$, $p < 0.001$) and soil total nitrogen (STN) (0–10 cm: $F = 31.93$, $p < 0.001$; 10–20 cm: $F = 155.63$, $p < 0.001$) decreased significantly from the large gap to the forest understory, and they decreased significantly as the elevation increased (;). There was a significant interaction effect between elevation and gap size on the soil C/N ratio (0–10 cm: $F = 2.36$, $p = 0.042$; 10–20 cm: $F = 6.47$, $p < 0.001$) (Table 5). In conclusion, improvements in understory illumination mainly come from canopy gap openings, and elevation has a relatively broader influence on environmental factors than the forest gap size.

**Table 5.** Two-way ANOVAs for the effects of elevation and gap size on environmental factors. Significant effects ($p < 0.05$) are bolded. PPFD: photosynthetic photon flux density; DAMT: annual average temperature difference between day and night; STC: soil total carbon; STN: soil total nitrogen.

| | Elevation | | Gap Size | | Elevation × Size | |
|---|---|---|---|---|---|---|
| | **F** | *p* | **F** | *p* | **F** | *p* |
| **PPFD** (µ mol m$^{-2}$ s$^{-1}$) | | | | | | |
| Cloudless | 0.50 | 0.485 | 1.49 | 0.229 | 0.75 | 0.393 |
| Overcast | 0.68 | 0.615 | 27.57 | **<0.001** | 0.27 | 0.971 |
| **Air Temperature** (°C) | | | | | | |
| Annual mean temperature | 134.52 | **<0.001** | 13.59 | **<0.001** | 5.07 | **<0.001** |
| DAMT | 139.05 | **<0.001** | 510.80 | **<0.001** | 50.97 | **<0.001** |
| Temperature in growing season | 151.53 | **<0.001** | 2.09 | 0.141 | 5.87 | **<0.001** |
| Temperature in non-growing season | 44.82 | **<0.001** | 85.31 | **<0.001** | 39.98 | **<0.001** |
| **Air Humidity** (%) | | | | | | |
| Annual mean humidity | 7.39 | **<0.001** | 51.40 | **<0.001** | 15.15 | **<0.001** |
| **Soil property** | | | | | | |
| Soil temperature (°C) 0–10 cm | 127.26 | **<0.001** | 0.09 | 0.914 | 0.51 | 0.836 |
| 10–20 cm | 106.56 | **<0.001** | 1.86 | 0.173 | 0.82 | 0.588 |
| STC (%) 0–10 cm | 11.68 | **<0.001** | 8.54 | **0.002** | 2.33 | **0.044** |
| 10–20 cm | 371.40 | **<0.001** | 298.50 | **<0.001** | 21.80 | **<0.001** |
| STN (%) 0–10 cm | 31.93 | **<0.001** | 12.23 | **<0.001** | 1.17 | 0.347 |
| 10–20 cm | 155.63 | **<0.001** | 134.36 | **<0.001** | 22.49 | **<0.001** |
| Soil C/N ratio (%) 0–10 cm | 2.48 | 0.065 | 1.68 | 0.203 | 2.36 | **0.042** |
| 10–20 cm | 26.16 | **<0.001** | 29.73 | **<0.001** | 6.47 | **<0.001** |

*3.3. Relationships between Regeneration Density and Environmental Factors*

A correlation analysis revealed the relationships between the regeneration densities of the three tree species and environmental factors in different life history stages (Table 6). For *Abies* seedlings and saplings, there were significant and positive correlations between regeneration density and the PPFD, AMT, ST, STC, and STN. For *Abies* small trees, DAMT and air humidity (AH) had significant and negative correlations with regeneration density. The other two species (*Acer* and *Betula*) showed similar relationships between regeneration density and environmental factors for the three life history stages (Table 6), demonstrating that *Acer* and *Betula* probably have similar life habits. Compared with the sapling regeneration density of *Abies*, the sapling regeneration densities of *Acer* and *Betula* had more significant and negative correlations with AH.

In total, 59.57% of the variance was represented by the plane (Figure 5A). The first dimension (Dim 1) explained 41.2% of the total variance, mainly reflecting species regeneration density, AH, and soil nutrient contents. The regeneration density values in the seedling, sapling, and small-tree stages, the STC, and the STN were the most positively correlated with the first dimension, and the AH and soil C/N ratio (10–20 cm) were negatively correlated with the first dimension (Figure 5B). The gap size had significantly different coordinates from 0 in the first dimension. The confidence ellipses had no overlaps among gaps of various sizes. The ellipses of large gaps were far from those of small and medium gaps. The second dimension (Dim 2) explained 18.4% of the total variance, mainly representing air temperature. The NGST and DAMT were positively correlated with the second dimension. For the elevation variable, the species regeneration densities were divergent among different elevation levels.

**Table 6.** The relationship between environmental factors and species regeneration density in different life stages. Significant relationships ($p < 0.05$) are bolded. PPFD_C = photosynthetic photon flux density on cloudless days; PPFD_O = photosynthetic photon flux density on overcast days; AMT = air annual mean temperature; DAMT = annual average temperature difference between day and night; GST = annual mean temperature of growing season; NGST = annual mean temperature of non-growing season; AH = air relative humidity; ST10 = soil temperature (0–10 cm); ST20 = soil temperature (10–20 cm); SC10 = soil total C (0–10 cm); SC20 = soil total C (10–20 cm); SN10 = soil total N (0–10 cm); SN20 = soil total N (10–20 cm); SCN10 = soil C/N ratio (0–10 cm); SCN20 = soil C/N ratio (10–20 cm).

| Environmental Variables | *Abies faxoniana* | | | | | | | | | | | | *Betula utilis* | | | | | | | | | | | | *Acer maximowiczii* | | | | | | | | | | | |
| | Seedling | | Sapling | | Small Tree | | Seedling | | Sapling | | Small Tree | | Seedling | | Sapling | | Small Tree | | Seedling | | Sapling | | Small Tree | |
| | r | p | r | p | r | p | r | p | r | p | r | p | r | p | r | p | r | p | r | p | r | p | r | p |
| PPFD_O | 0.87 | **<0.001** | 0.78 | **<0.001** | 0.58 | **<0.001** | 0.75 | **<0.001** | 0.52 | **<0.001** | 0.43 | **0.001** | 0.76 | **<0.001** | 0.55 | **<0.001** | 0.63 | **<0.001** |
| PPFD_C | 0.86 | **<0.001** | 0.65 | **<0.001** | 0.72 | **<0.001** | 0.78 | **<0.001** | 0.61 | **<0.001** | 0.53 | **<0.001** | 0.83 | **<0.001** | 0.60 | **<0.001** | 0.66 | **<0.001** |
| AMT | 0.35 | **0.006** | 0.43 | **0.001** | 0.26 | **0.043** | 0.39 | **0.002** | 0.41 | **0.001** | 0.40 | **0.002** | 0.32 | **0.011** | 0.47 | **<0.001** | 0.44 | **<0.001** |
| DAMT | −0.20 | 0.122 | 0.18 | 0.168 | −0.45 | **<0.001** | −0.19 | 0.139 | −0.26 | **0.047** | −0.27 | **0.035** | −0.30 | **0.019** | −0.17 | 0.194 | −0.20 | 0.126 |
| GST | 0.16 | 0.208 | 0.23 | 0.073 | 0.16 | 0.219 | 0.30 | **0.019** | 0.37 | **0.004** | 0.33 | **0.009** | 0.15 | 0.250 | 0.39 | **0.002** | 0.30 | **0.020** |
| NGST | −0.14 | 0.272 | 0.06 | 0.672 | −0.30 | **0.020** | −0.13 | 0.320 | −0.24 | 0.068 | −0.33 | **0.010** | −0.21 | 0.101 | −0.19 | 0.145 | −0.23 | 0.079 |
| AH | −0.15 | 0.265 | −0.13 | 0.331 | −0.31 | **0.016** | −0.24 | 0.065 | −0.33 | **0.010** | −0.41 | **0.001** | −0.32 | **0.014** | −0.37 | **0.003** | −0.33 | **0.009** |
| ST10 | 0.34 | **0.007** | 0.39 | **0.002** | 0.27 | **0.039** | 0.43 | **0.001** | 0.47 | **<0.001** | 0.45 | **<0.001** | 0.30 | **0.019** | 0.50 | **<0.001** | 0.45 | **<0.001** |
| ST20 | 0.41 | **0.001** | 0.40 | **0.001** | 0.35 | **0.006** | 0.51 | **<0.001** | 0.55 | **<0.001** | 0.52 | **<0.001** | 0.37 | **0.004** | 0.57 | **<0.001** | 0.52 | **<0.001** |
| SC10 | 0.53 | **<0.001** | 0.29 | **0.023** | 0.56 | **<0.001** | 0.64 | **<0.001** | 0.71 | **<0.001** | 0.72 | **<0.001** | 0.51 | **<0.001** | 0.69 | **<0.001** | 0.67 | **<0.001** |
| SC20 | 0.57 | **<0.001** | 0.31 | **0.014** | 0.51 | **<0.001** | 0.64 | **<0.001** | 0.58 | **<0.001** | 0.44 | **<0.001** | 0.49 | **<0.001** | 0.54 | **<0.001** | 0.50 | **<0.001** |
| SN10 | 0.67 | **<0.001** | 0.50 | **<0.001** | 0.63 | **<0.001** | 0.76 | **<0.001** | 0.74 | **<0.001** | 0.67 | **<0.001** | 0.62 | **<0.001** | 0.72 | **<0.001** | 0.67 | **<0.001** |
| SN20 | 0.62 | **<0.001** | 0.46 | **<0.001** | 0.52 | **<0.001** | 0.64 | **<0.001** | 0.53 | **<0.001** | 0.43 | **0.001** | 0.56 | **<0.001** | 0.53 | **<0.001** | 0.53 | **<0.001** |
| SCN10 | 0.01 | 0.939 | −0.15 | 0.241 | 0.13 | 0.340 | 0.07 | 0.609 | 0.21 | 0.107 | 0.29 | **0.025** | 0.02 | 0.888 | 0.19 | 0.143 | 0.20 | 0.122 |
| SCN20 | −0.32 | **0.013** | −0.42 | **0.001** | −0.17 | 0.196 | −0.23 | 0.079 | −0.11 | 0.413 | −0.12 | 0.379 | −0.28 | **0.029** | −0.17 | 0.199 | −0.22 | 0.091 |

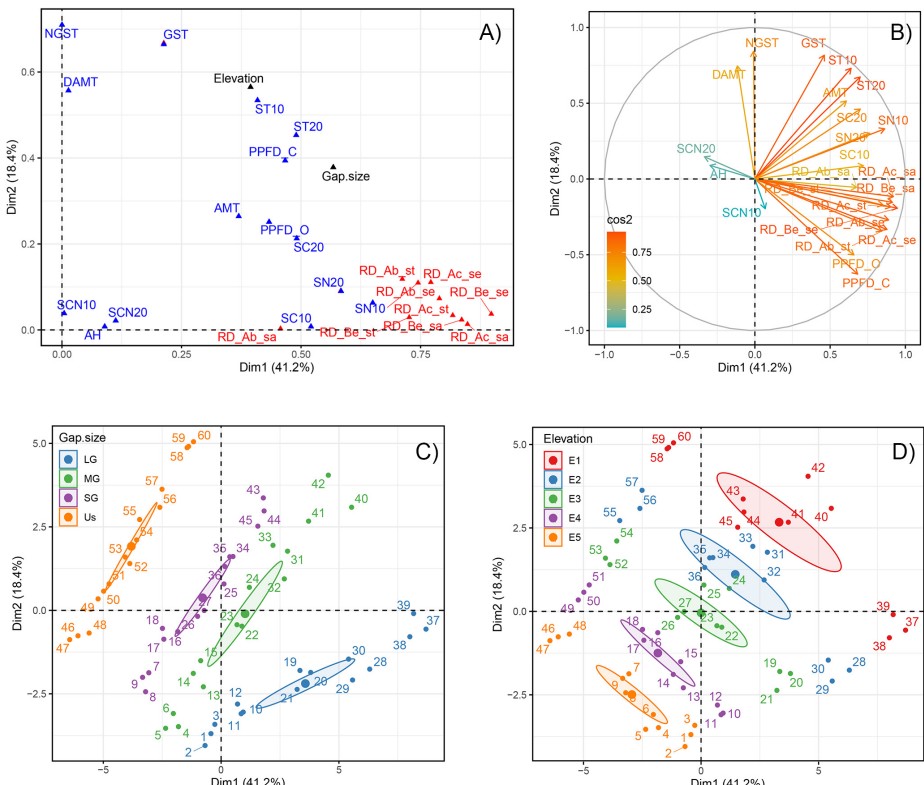

**Figure 5.** Factor analysis of mixed data (FAMD) results for the relationships between regeneration density and environmental factors in the first two dimensions. (**A**) All quantitative and qualitative variables; (**B**) quantitative variables (correlation circle graph). The cos2 value (i.e., the squared cosine value) represents the importance of a variable, and a higher cos2 indicates a larger contribution to the principal components; (**C**) multiple categorical variables are colored separately for gap size (LG: large gap; MG: medium gap; SG: small gap; Us: understory); (**D**) multiple categorical variables are colored separately for elevation (E1: 3000–3100 m; E2: 3100–3200 m; E$_3$: 3200–3300 m; E$_4$: 3300–3400 m; E$_5$: 3400–3500 m). RD_Ab_se = *Abies* seedling regeneration density; RD_Ab_sa = *Abies* sapling regeneration density; RD_Ab_st = *Abies* small-tree regeneration density; RD_Be_se = *Betula* seedling regeneration density; RD_Be_sa = *Betula* sapling regeneration density; RD_Be_st = *Betula* small-tree regeneration density; RD_Ac_se = *Acer* seedling regeneration density; RD_Ac_sa = *Acer* sapling regeneration density; RD_ Ac _st = *Acer* small-tree regeneration density; PPFD_C = photosynthetic photon flux density on cloudless days; PPFD_O = photosynthetic photon flux density on overcast days; AMT = air annual mean temperature; DAMT = annual average temperature difference between day and night; GST = annual mean temperature of growing season; NGST = annual mean temperature of non-growing season; AH = air relative humidity; ST10 = soil temperature (0–10 cm); ST20 = soil temperature (10–20 cm); SC10 = soil total C (0–10 cm); SC20 = soil total C (10–20 cm); SN10 = soil total N (0–10 cm); SN20 = soil total N (10–20 cm); SCN10 = soil C/N ratio (0–10 cm); SCN20 = soil C/N ratio (10–20 cm).

## 4. Discussion

### 4.1. Effect of Gap Size and Elevation on the Regeneration Niche for Species Coexistence

The niche breadth in the sapling stage was the narrowest among the different life history stages (seedlings, saplings, and small trees) of the three tree species (Table 1). This suggests that the niche requirements of saplings are more restrictive. The regeneration niches of *Abies* and *Betula* narrowed down, whereas the regeneration niche of *Acer* broadened during ontogeny. For *Abies*, the regeneration niche from the seedling stage to the sapling stage was wider than the regeneration niche from the sapling to small-tree stages (Table 1). This shows that the conditions for the successful establishment of small trees are more stringent than the conditions for the successful formation of saplings [16]. For *Betula*,

the regeneration niche breadths were similar during different ontogeny stages. For *Acer*, the formation of saplings is more difficult than other life history stages during ontogeny.

The forest gap size can affect the microenvironments in gaps, which has impacts on species regeneration [34,35]. The PPFD was mainly affected by the gap size on overcast days during the growing season (Table 2) and has a positive and significant correlation with the regeneration densities of seedlings, saplings, and small trees in subalpine gaps (Table 6). For *Abies*, a common shade-tolerant species, the regeneration density and niche were less affected by gap size. Small gaps with lower PPFD values provided more superior habitats for *Abies* saplings [36]. To sustain growth, light demands also increase in the small-tree life stage for *Abies* [4,19]. Some studies suggest that the higher light found in the center of small gaps may allow for shade-tolerant species to be established [2,7]. We speculate that previous regeneration might contribute to the survival of small *Abies* trees in small gaps because several *Abies* saplings may have been established in the understory and will grow rapidly once gaps are formed. Therefore, *Abies* can regenerate steadily as the canopy dynamic changes.

Several studies in the conifer forest reported that the GST was not significantly affected by gap size [1,37,38]. There was no significant difference in the GST among different gap sizes, whereas it was significantly affected by elevation (Table 6). Our study showed that the effects of elevation and gap size interaction significantly influenced the GST. The GST was positively and significantly correlated with the regeneration densities of *Betula* seedlings, saplings, and small trees and *Acer* saplings and small trees. In medium and large gaps, *Betula* regeneration density was significantly higher than *Acer* regeneration density, and the regeneration niche breadth of *Betula* was wider than the regeneration niche breadth of *Acer*. Thus, it shows that *Betula* has greater competitiveness and a more effective regeneration strategy in the seedling and sapling life stages than *Acer* in a suitable microsite.

The STC, STN, and C/N ratios were positively and significantly affected by gap size and the interaction of elevation and gap size. Soil nutrients had positive and significant correlations with the regeneration densities of the seedlings, saplings, and small trees of the three tree species. For sapling regeneration, *Betula* and *Acer* were more significantly related to fertile soil nutrients in large gaps than *Abies* [1]. Therefore, these findings suggest that during early succession, the natural regeneration of pioneer tree species, such as *Betula*, was restricted by soil nutrition in addition to solar radiation and demands large gaps in the subalpine coniferous forest. Sanchez et al. (2013) suggested that the soil characteristics of the altitudinal gradient determine the distribution of plant patterns [12]. In our study, almost all of the underground and below-ground environmental factors were significantly affected by elevation except for the PPFD. For the regeneration of *Betula*, the PPFD, air temperature, AH, and soil properties were significantly correlated with the regeneration density of saplings. These facts explain why the *Abies* saplings had a regeneration advantage in gaps with increasing elevation, whereas the regeneration of *Betula* saplings was more dominant at lower elevation gradients. Furthermore, the effect of elevation on the regeneration of the observed species was more obvious in large gaps. For the canopy opening area, Denslow (1987) pointed out that the environmental factors in gaps would be more sensitive to harsh weather conditions without the protection of the canopy [39]. Considering the extensive influence of elevation on environmental factors, the filtration capacity from the elevation for the establishment and regeneration of plants may be greater as the gap size increases.

*4.2. Community Assembly Processes along Gap and Elevation Gradients*

The coexistence of species in communities is due to different ecological strategies for occupying different niches. Environmental heterogeneity affecting niche space is a primary factor in species coexistence [40,41]. Bergholz et al. (2017) showed that environmental heterogeneity resulting from environmental filtering within microhabitats had a major influence on community assembly [42]. Microhabitat heterogeneity is controlled by the characteristics of forest gaps, which affect tree regeneration. Our study demonstrates that the sapling regeneration of the shade-intolerant species *Betula* dominated in large and

medium gaps, and the regeneration advantage could be extended to the small-tree stage in large gaps at lower elevation gradients (3000–3300 m).

The three tree species of the subalpine coniferous forest in the study area have different regeneration ecologies. The conifer *Abies* is a long-living climax species and could endure a competitive disadvantage with a lower regeneration density in relatively resource-rich gaps in the early stages of succession [18]. As mentioned by Jin et al. (2018), light radiation limits the regeneration of shade-tolerant species in large gaps [19]. In addition, the slow allocation to height growth of *Abies* results in tall and hard stems, which appears to be a long-term competitive strategy to tolerate shading [24]. However, *Betula* is a shade-intolerant species that grows fast under full sunlight, resulting in tall young trees with slender trunks and narrow crowns. *Betula* grows rapidly under full sunlight and becomes a pioneer species in the early stages of succession. This competitive strategy results in higher rates of mortality from the sapling to small-tree stages [36]. With the increase in continuous canopy cover in later stages of succession, *Betula* is replaced by shade-tolerant species such as *Abies*. Thus, *Abies* is more representative of k-selected species compared to r-selected species like *Betula* [36]. For deciduous *Acer* regeneration, it survives as a fugitive with the lowest sapling and small-tree regeneration densities. This species seems to avoid competition with *Abies* and *Betula* and adapts to a relatively narrow niche even though the habitat is unsuitable (e.g., large gaps and medium gaps) [18]. Light radiation limits the regeneration of shade-tolerant species in large gaps, as mentioned by Jin et al. (2018) [19]. Large gaps may provide excellent opportunities and places for *Betula* and *Acer* to become canopy trees [43,44]. Environmental conditions, including soil conditions and light intensity controlled by topographic features, affect the sapling regeneration of these three tree species and the processes of community assembly.

*4.3. Forest Management Implications*

We explored the effects of forest gap size on the regeneration density of the three tree species along an elevation gradient. These species showed different regeneration and coexistence ecology strategies. Our study revealed that large gaps could enlarge the effects of elevation on a species' regeneration density. Moreover, a related study revealed that elevation shifts can locally reshape the competitiveness of species [45]. Thus, low-intensity canopy strategies could be applied in future forest-management plans if the dominance regeneration of *Abies* is maintained along the elevation gradients. Broadleaf tree species, especially *Betula* and *Acer*, play a crucial role in forming "colored forests" in autumn [25]. In addition, the long-term monoculture of an *Abies* plantation could lead to increased vulnerability to disease, pest disturbances, reduced soil fertility, and other potential ecological problems [7]. Thus, large or medium gaps could promote the regeneration of broadleaf species and improve ecological function and sustainability. Therefore, we could take advantage of the differences in the regeneration niche among different tree species to select suitable tree mixtures for forest restoration and management.

**5. Conclusions**

This study revealed that the regeneration densities of the three tree species at the seedlings, sapling, and small-trees stages all increased with an increase in the gap size. The regeneration density and niche of *Abies* were less affected by gap size due to a tolerance for shade. Therefore, small gaps with lower PPFD values were conducive to the regeneration of *Abies* saplings. The shade-intolerant species *Betula* has stronger competitiveness and a more effective regeneration strategy in medium and large gaps. Gap size could positively enlarge the effects of elevation on the regeneration densities of the three species. The PPFD was significantly affected by gap size, whereas the habitat temperature was significantly affected by elevation. The connection between the species' regeneration densities and habitat conditions reflect the selective pre-emption of environmental resources of different species in different life stages, which is due to differences in the regeneration niche, affecting species coexistence and community assembly. This study showed clear evidence of species

regeneration and coexistence in a potential ecological mechanism with typical habitat gradients in a subalpine coniferous forest. It also provided a basis for forest management and sustainable development.

**Supplementary Materials:** The following supporting information can be downloaded at: https://www.mdpi.com/article/10.3390/f14102099/s1, Table S1: Mean environmental factors in different gap sizes ranging from 3000 to 3500 m.

**Author Contributions:** W.H. and L.C. designed the study, analyzed the data, and wrote the manuscript; D.L. and J.L. organized and analyzed the data; G.G.W. and G.L. modified the article structure. All authors have read and agreed to the published version of the manuscript.

**Funding:** This study was supported by the Second Tibetan Plateau Scientific Expedition and Research Program (STEP) (grant no. 2019QZKK0402), the National Natural Science Foundation of China (grant no. 32301499), the National Key Research and Development Program of China (2016YFC0502102), the Jiangsu Provincial Double-Innovation Doctor Program (JSSCBS20210456), and the Startup Foundation for Introducing Talent of Nanjing University of Information Science & Technology (NUIST) (grant No. 2021r098 and 003319).

**Data Availability Statement:** The data in this study are available from the authors upon request.

**Acknowledgments:** The authors thank all those who helped us complete this research. We gratefully acknowledge financial support from the China Scholarship Council.

**Conflicts of Interest:** The authors declare no conflict of interest.

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
