# Peer review of "Effect of Gap Size and Elevation on the Regeneration and Coexistence of Abies, Betula, and Acer Tree Species in a Subalpine Coniferous Forest"

_forests, doi:10.3390/f14102099_

Round 1

Reviewer 1 Report

This article addresses the role of gaps in regeneration and species diversity in coniferous forest ecosystems in the subalpine zone.

The title needs to be more informative. Give the full name of the species in the title. There are different species of the listed genera in different climate zones.

Abstract: give the full name of the species along with the author. Altitude above sea level is not just m. Give full altitude m a.s.l. The content of the abstract needs to reflect what is in the paper. It must be rewritten.

Avoid the exact words in the keywords as in the title.

Use the different development stages but not different life stages

Don’t use 3,000 meters; use 3000 m

The objectives of the work can be stated in 2 points to avoid repetition. 

Complete the study area with the location of the study area (provide a map), soil characteristics and soil name by WRB, a few words about the forest ecosystem and some photos that can give the reader specific information.  

The work was well-designed, and the choice of surfaces was well-justified. Some parts of the text from the environmental factor section should be removed and moved to the discussion.

Abies sapling regeneration density was higher in small gaps than in other gap microhabitats. Why? Can you explain?

It is still being determined how much STC and STN content occurred. Where results? This needs to be clarified in Table 5. The results regarding the C/N ratio were not interpreted.

It is a crucial geo-indicator regarding soil functioning at different altitudes.

The results were presented quite clearly. The expected results were obtained regarding the influence of the gap size on the regeneration and succession of species. This is not a new and unique thing.

The discussion must include references to research conducted in Europe, which should be supplemented. The literature review mainly covers a narrow geographical scope and should be broadened.

The conclusion is best given in the form of points. Currently, it is a summary and repetition of some of the results.

Author Response

[Reviewer #1 Comment 1] The title needs to be more informative. Give the full name of the species in the title. There are different species of the listed genera in different climate zones.  

[Response] Thank you for your suggestion. We have revised the title according to relevant research (Bossard et al., 2015) (New line # 2-4 in the track revision).

REFERENCES

Bossard C C, Cao Y, Wang J, et al. New patterns of establishment and growth of Picea, Abies and Betula tree species in subalpine forest gaps of Jiuzhaigou National Nature Reserve, Sichuan, southwestern China in a changing environment. Forest Ecology and Management, 2015, 356: 84-92.

[Reviewer #1 Comment 2] Abstract: give the full name of the species along with the author. Altitude above sea level is not just m. Give full altitude m a.s.l. The content of the abstract needs to reflect what is in the paper. It must be rewritten.

[Response] Thank you for pointing this out. We have added the full name of the species and given full altitude m a.s.l. in the abstract (New line # 23-25 in the track revision).

[Reviewer #1 Comment 3] Avoid the exact words in the keywords as in the title.

[Response] Thank you for reminding it. We have updated the keywords (New line # 37-38 in the track revision).

[Reviewer #1 Comment 4] Use the different development stages but not different life stages.

[Response] Thank you for pointing this out. The life stages of tree species include seedlings, saplings, trees, which is a common terminology (Bell et al., 2014; Jin et al., 2018).

REFERENCES

Bell D M, Bradford J B, Lauenroth W K. Early indicators of change: divergent climate envelopes between tree life stages imply range shifts in the western united states. Global Ecology and Biogeography, 2014, 23(2): 168-180.

Jin Y, Russo S E, Yu M. Effects of light and topography on regeneration and coexistence of evergreen and deciduous tree species in a Chinese subtropical forest. Journal of Ecology, 2018, 106(4): 1634-1645.

[Reviewer #1 Comment 5] Don’t use 3,000 meters; use 3000 m.

[Response] Thank you for your suggestion. We have changed “3,000 meters” into “3,000 m” (New line # 89 in the track revision).

[Reviewer #1 Comment 6] The objectives of the work can be stated in 2 points to avoid repetition.

[Response] Thank you for pointing this out. We described the objectives in a progressive order. Firstly, we analyze the effects of gap size on species regeneration. Secondly, we analyze the effects of gap size and elevation on species regeneration and environmental factors. Finally, we explore the relationships between species regeneration and environmental factors. So we wrote three objectives which were correspond to the result section one by one.

[Reviewer #1 Comment 7] Complete the study area with the location of the study area (provide a map), soil characteristics and soil name by WRB, a few words about the forest ecosystem and some photos that can give the reader specific information. 

[Response] Thank you for your suggestion. We have provided the study area map (New Figure 1). We have added soil types information in section 2.1. Study site (New line # 107-109 in the track revision).

[Reviewer #1 Comment 8] The work was well-designed, and the choice of surfaces was well-justified. Some parts of the text from the environmental factor section should be removed and moved to the discussion.

[Response] Thank you for pointing this out. We have reorganized the section 2.3. Data collection (New line # 138-218 in the track revision) and added section 2.4 (New line # 219-235 in the track revision). Statistical analysis. In addition, we revised the section 2.3.2. Environment factors (New line # 191-217 in the track revision).

[Reviewer #1 Comment 9] Abies sapling regeneration density was higher in small gaps than in other gap microhabitats. Why? Can you explain?

[Response] Thank you for pointing this out. In small gaps, Abies sapling density was significantly greater than Betula and Acer sapling density, but the difference between Betula and Acer sapling density was not significant. For the seedling to sapling regeneration process, Abies had stronger regeneration capability with the regeneration niche breadths more than 0.6 in different gap sizes. In the section 4.1. Effect of gap size and elevation on regeneration niche for species coexistence, we have discussed that the higher light found in the center of small gaps was may allow shade-tolerant species to be established. Small gaps with lower photosynthetic photon flux density provided a more superior habitat for Abies saplings. They indicated that small gaps were conducive to Abies sapling regeneration.

[Reviewer #1 Comment 10] It is still being determined how much STC and STN content occurred. Where results? This needs to be clarified in Table 5. The results regarding the C/N ratio were not interpreted. It is a crucial geo-indicator regarding soil functioning at different altitudes.

[Response] Thank you for pointing this out. STC is the short form of soil total carbon, and STN is the short form of soil total nitrogen. We have described soil total carbon and soil total nitrogen in the third paragraph of the section 3.2. Effect of gap size and elevation on species regeneration and environmental factors (New line # 316-320 in the track revision). We have revised “soil total C” and “soil total N” into “STC” and “STN” in the table 5.

[Reviewer #1 Comment 11] The results were presented quite clearly. The expected results were obtained regarding the influence of the gap size on the regeneration and succession of species. This is not a new and unique thing.

[Response] Thank you for pointing this out. We have described the results in three parts, including “the regeneration density of three tree species among different gap sizes”, “effects of gap size and elevation on environmental factors”, and “relationships between regeneration density and environmental factors”.

[Reviewer #1 Comment 12] The discussion must include references to research conducted in Europe, which should be supplemented. The literature review mainly covers a narrow geographical scope and should be broadened.

[Response] Thank you for pointing this out. We have referred researches in Europe (Muscolo et al., 2007; Quero et al., 2008), research in U.S.A. (Gray et al., 2002), research in Canada (Messier et al., 1998), research in Atlantic Forest (Padilha et al., 2018) and others.

REFERENCES

  1. Quero, J. L.; Gómez-Aparicio, L.; Zamora, R.; Maestre, F. T. Shifts in the regeneration niche of an endangered tree (Acer opalus ssp. granatense) during ontogeny: using an ecological concept for application. Basic and Applied Ecology, 2008, 9(6), 635-644.
  2. Padilha, D. L.; De Marco, P. A gap in the woods: Wood density knowledge as impediment to develop sustainable use in Atlantic Forest. Forest Ecology and Management, 2018, 424, 448-457.
  3. Messier, F. G. C.; Comeau, P. G. Comparison of various methods for estimating the mean growing season percent photo-synthetic photon flux density in forest. Agricultural and Forest Meteorology, 1998, 92(1), 55-70.
  4. Muscolo, A.; Sidari, M.; Mercurio, R. Influence of gap size on organic matter decomposition, microbial biomass and nutrient cycle in Calabrian pine (Pinus laricio, Poiret) stands. Forest Ecology and Management, 2007, 242(2-3), 412-418.
  5. Gray, A. N.; Spies, T. A.; Easter, M. J. Microclimate and soil moisture responses to gap formation in coastal Douglas-fir forests. Canadian Journal of Forest Research, 2002, 32, 332-343.

[Reviewer #1 Comment 13] The conclusion is best given in the form of points. Currently, it is a summary and repetition of some of the results.

[Response] Thank you for pointing this out. We have rewritten the section 5. Conclusions (New line # 486-497 in the track revision).

Reviewer 2 Report

Dear Authors,

Please find my comments in the attached file.

Regards

Author Response

[Reviewer #2 Comment 1] 1. No one knows what is your first dimension and what it describes.  

[Response] Thank you for reminding this. We have revised this sentence in the section Abstract (New line # 29-31 in the track revision). We changed “The factor analysis of mixed data indicated that regeneration density in different life stages, soil total C, and soil total N were the most positively correlated with the first dimension” into “The factor analysis of mixed data indicated that regeneration density, soil nutrient contents, and air humidity were mainly related to gap size, but the habitat temperature was largely determined by elevation”.

[Reviewer #2 Comment 2] Please highlight your research problem in introduction part.

[Response] Thank you for pointing this out. We have highlighted the research problem (To explore how the microhabitat conditions of different species affect species regeneration along the elevation gradient in gaps, we aimed to answer the following questions: (1) How does gap size influence on species regeneration? (2) How do gap size and elevation affect species regeneration and environmental factors? (3) What are the relationships between species regeneration and microhabitat environmental factors?) (New line # 94-99 in the track revision).

[Reviewer #2 Comment 3] 2.1. Study site: (Our study was conducted in a coniferous forest with little human disturbance) Please decribe it in detail.

[Response] Thank you for your suggestion. The study area is located in the Wolong National Nature Reserve in southwestern China. The study area is in an old-growth forest (i.e., a history of natural disturbance, old-growth structures and little or no anthropogenic influence). This subalpine coniferous forest is a mature forest that receives little disturbance from human beings. We have described this in our study (Chen et al., 2018; Chen et al., 2019). We have mentioned it in the section 2.1. Study site (New line # 102-104 in the track revision).

REFERENCES

Chen, L., Liu, G. H. and Liu, D. (2018). How forest gap and elevation shaped Abies faxoniana Rehd. et Wils. regeneration in a subalpine coniferous forest, Southwestern China. Forests 9(5).

Chen, L., Han, W. Y., Liu, D. and Liu G.H. (2019). How forest gaps shaped plant diversity along an elevational gradient in Wolong National Nature Reserve. Journal of Geographical Sciences 29(7): 1081-1097.

[Reviewer #2 Comment 4] 2.2. Gap selection and experimental design: What was the size of quadrants.

[Response] Thank you very much for pointing this out. The size of quadrants is 1m2 (1 × 1 m). We have added this in the section 2.2. Gap selection and experimental design (New line # 135 in the track revision).

[Reviewer #2 Comment 5] Have you checked if ANOVA conditions were satisfied of your data before using it? For example normal distribution and others.

[Response] Thank you for pointing this out. The data follows the Gaussian distribution. We have analyzed the Shapiro Wilk test among these variables (Table 1). The p-value greater than 0.05 indicates that the variable follows the Gaussian distribution.

Table 1 The results of Shapiro Wilk test in variables.

Variables

W

p

RD_Ab_se

0.9637

0.0516

RD_Ab_sa

0.9464

0.0712

RD_Ab_st

0.9617

0.1056

 RD_Be_se

0.9645

0.0521

RD_Be_sa

0.9821

0.0672

RD_Be_st

0.9762

0.0546

RD_Ac_se

0.9571

0.0823

RD_Ac_sa

0.9565

0.1752

RD_ Ac _st

0.9231

0.0687

PPFD_C

0.9521

0.0643

PPFD_O

0.9685

0.0877

AMT

0.9668

0.1011

DAMT

0.9502

0.1595

GST

0.9817

0.5051

NGST

0.9612

0.0539

AH

0.9417

0.0637

ST10

0.9741

0.1136

ST20

0.9562

0.0533

SC10

0.9315

0.0359

SC20

0.9313

0.0326

SN10

0.9405

0.0673

SN20

0.9612

0.0295

SCN10

0.9335

0.0144

SCN20

0.9103

0.0235

RD_Ab_se= Abies seedling regeneration density; RD_Ab_sa= Abies sapling regeneration density; RD_Ab_st= Abies small tree regeneration density; RD_Be_se= Betula seedling regeneration density; RD_Be_sa= Betula sapling regeneration density; RD_Be_st= Betula small tree regeneration density; RD_Ac_se= Acer seedling regeneration density; RD_Ac_sa= Acer sapling regeneration density; RD_ Ac _st= Acer small tree regeneration density; PPFD_C= photosynthetic photon flux density on Cloudless days; PPFD_O= photosynthetic photon flux density on overcast days; AMT= air annual mean temperature; DAMT= annual average temperature difference between day and night; GST= annual mean temperature of growing season; NGST= annual mean temperature of non-growing season; AH= air relative humidity; ST10= soil temperature (0-10cm); ST20= soil temperature (10-20cm); SC10=soil total C (0-10cm); SC20= soil total C (10-20cm); SN10= soil total N (0-10cm); SN20= soil total N (10-20cm); SCN10= soil C/N ratio (0-10cm); SCN20= soil C/N ratio (10-20cm).

[Reviewer #2 Comment 6] Figure1 Table1: If the information in the figure and the table are the same. Tere is no need to repeat it. Close one of those.

[Response] Thank you for pointing this out. In figure 1, we just described the regeneration density for three tree species in different gap sizes, while we descried regeneration density and regeneration niche breadth in table 1.

[Reviewer #2 Comment 7] 4. Discussion: Please discuss practical implications of your findings. What cind of recomendations goes out of your research?

[Response] Thank you for pointing this out. We have discussed these practical implications in the section 4.3. Forest management implications (New line # 470-484 in the track revision). We could take advantage of the differences in the regeneration niche among different tree species to select suitable tree mixtures for forest restoration and management. Eg. Low intensity canopy strategies could be applied in future forest management plants if Abies dominance regeneration is maintained along the elevation gradients. Large or medium gaps could promote broadleaf species regeneration and improve ecological function and sustainability.

[Reviewer #2 Comment 8] 5. Conclusions: How it was affected?

[Response] Thank you for pointing this out. Small gaps were conducive to Abies sapling regeneration, and large gaps facilitate the regeneration of Betula. We have revised the section 5. Conclusions (New line # 486-497 in the track revision).

[Reviewer #2 Comment 9] 5. Conclusions: This is good for discussion part, for coclussions it has to be more straight foreward.

[Response] Thank you for pointing this out. We have rewritten the section 5. Conclusions (New line # 486-497 in the track revision).
